# Towards Understanding Learning from Human Interventions

## Abstract

As AI systems are deployed in real-world environments, they inevitably make mistakes where human interventions could provide valuable corrective feedback. However, many of the optimality assumptions made by existing methods for learning from interventions are invalid or unrealistic when measured against how humans actually intervene in reality. We conduct a deeper analysis with intervention data from real human users, revealing that humans often intervene sub-optimally in both the timing and execution of interventions, often acting when they perceive the agent's progress to stagnate. Building on these insights, we show that the current methods of simulating human interventions, and the corresponding methods to learn from these interventions, do not accurately capture the behavior modes of human users in practice. Based on these insights, we introduce an improved approximate model of human intervention that better captures this behavior, enabling accurate simulation benchmarking of learning algorithms and providing a more reliable signal to develop better algorithms in the future. As a start to building on these insights, we propose a simple algorithm that combines imitation learning and reinforcement learning with a regularization scheme to leverage corrections for exploration rather than directly making strong optimality assumptions. Our empirical evaluation on simulated robotic manipulation tasks demonstrates that our method improves task success by $\sim$52% and achieves $\sim$2x reduction in real-human effort on average as compared to baselines, marking a significant step towards scalable, human-interactive learning for robot manipulation.

## 1 Introduction

Even with state-of-the-art machine learning techniques, policies for sequential decision-making often struggle to generalize beyond their training distribution (Koh et al., 2020). For instance, in robotics, imitation learning (IL) from large datasets suffers from compounding errors due to covariate shift (Osa et al., 2018), while simulation-trained policies fail in the real world due to the simulation-reality dynamics gap (Peng et al., 2017). Similarly, large language models (LLMs) trained on web-scale text corpora frequently exhibit hallucinations on out-of-distribution (OOD) inputs (Ji et al., 2023; Kang et al., 2024) or in different test domains (Gururangan et al., 2020). For robust deployment of such systems, policies should adapt online from feedback (Ouyang et al., 2022; Ross et al., 2011) rather than relying solely on fixed offline datasets.

To address these distribution shifts, we can rely on humans as powerful sources of real-world feedback. Reinforcement learning from human feedback (RLHF) (Christiano et al., 2017; Ouyang et al., 2022) has proven to be a dominant tool for improving the performance of AI systems, including LLMs, where feedback is provided as binary preferences over alternative responses. For robots, human input on robot behavior could enable adaptation to novel settings and improved task success. However, existing methods (Torne et al., 2024; 2023; Bıyık et al., 2022; Biyik & Sadigh, 2018) typically rely on tele-operated demonstrations or binary feedback on states, which require expensive task executions or provide sparse supervision.

Instead, a more natural approach is to assume that humans will give feedback through corrections (Michael et al., 2019; Jiang et al., 2024; Liang et al., 2024), where humans correct the agent only when it gets stuck or makes mistakes. This type of feedback arises organically when people interact with AI systems, such as when they edit AI-generated code and images (Brooks et al., 2023),

or physically steering robots towards the goal (see Figure 1) (Bajcsy et al., 2018; Mandlekar et al., 2020). Learning from such corrections involves policy adaptation based on their implicit preference signals i.e. the human correction is preferred over the agent's current behavior. Methods leveraging human interventions for robot learning have used them as partial demonstrations (Liu et al., 2023; Mandlekar et al., 2020; Michael et al., 2019; Luo et al., 2024b) to imitate preferred behavior, or as reward shaping signals Knox & Stone (2009); Bajcsy et al. (2018); Xie et al. (2022); Luo et al. (2024a); Korkmaz & Biyik (2025) for RL, or as constraints (Lindner et al., 2022; Spencer et al., 2022; Ainsworth et al., 2019) over agent behavior.

In this work, we focus on learning from physical human interventions for robots. A key bottleneck is that we need a principled understanding of *when* and *how* humans intervene, both to guide our algorithmic design choices and evaluate these methods. As real-world training and evaluation is slow and expensive, we inevitably need grounded simulators of human behavior. Such models enable rigorous, repeatable evaluation and systematically guide algorithmic components (e.g., how to incorporate corrections, the exploration-safety trade off, robustness to noise) prior to deployment. For example, prior work models interventions as reactions to instantaneous action suboptimality (value gap between the optimal and policy action (Luo et al., 2024a)), or as Boltzmann rational (Bradley & Terry, 1952) functions proportional to expert action value (Bajcsy et al., 2018), or as probit choice model, intervening when the human's action

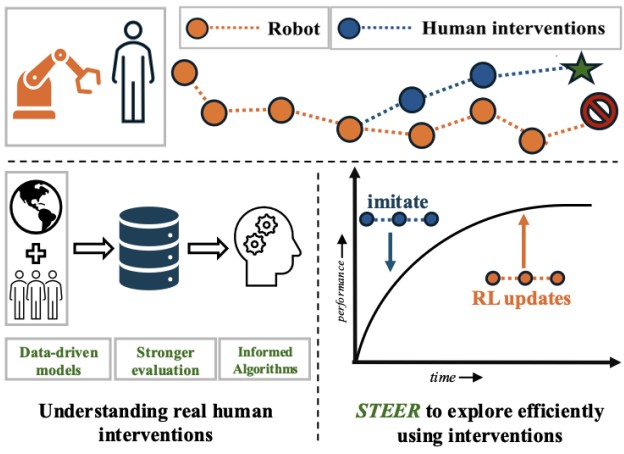

Figure 1: Current intervention learning approaches use incorrect human assumptions Korkmaz & Biyik (2025); Luo et al. (2024a) or inefficient training schemes for policy adaptation Luo et al. (2024b). Left: Real human intervention data and analysis, yields a progress based non-Markovian model with better alignment to observed behavior for realistic evaluation. Right: STEER - a hybrid BC+RL method that uses BC on human corrections to steer exploration, then fades to pure RL via a decaying weight, enabling faster learning with fewer interventions and robustness to noise.

considerably exceeds the robot's expected return (Korkmaz & Biyik, 2025). However, these formulations are insufficiently grounded in real world behavior.

In Section 4, we present a comprehensive analysis of real human interventions in robot manipulation and observe that rather than responding to instantaneous sub-optimality, humans intervene when the agent's progress over a short horizon falls below a threshold. This pattern aligns with cognitive models of caregiving, where intervention depends on the agent's ability over time rather than isolated states (Shachnai et al., 2025). Motivated by these findings, we propose a simple *stagnation-based intervention model* that aggregates progress over a past horizon and intervenes based on progress sub-optimality. Intuitively, a human monitors behavior over a window, and intervenes if improvement is insufficient. This temporal criterion produces a more realistic simulator with higher correlation to observed human behavior (Section 4.4). Critically, our analysis and model are grounded in real data, not hypothetical feedback assumptions, providing a stronger basis for evaluation.

Additionally, we observe that interventions are inherently suboptimal (Section 4.2): they are invalid as ground-truth demonstrations, or perfect reward signals, contrary to prior works (Liu et al., 2023; Mandlekar et al., 2020; Michael et al., 2019; Luo et al., 2024a;b). In Section 4.3, we show that human behavior diverges from the learner policy as training progresses, because as humans only partially observe the policy and task, their corrections can steer the robots towards valid yet conflicting solutions w.r.t the current policy behavior. Our key insight is to treat interventions for exploration, rather than optimal supervision targets. Consequently, interventions only bias exploration in training towards promising regions and RL can then recover optimal behavior from the (suboptimal) exploration data (Kostrikov et al., 2022; Levine et al., 2020; Chen et al., 2021). Prior works used corrections as off-policy data, but we find that this is prohibitively slow as compared to simply imitating these transitions with a maximum likelihood loss with a decaying weight. This proves to

be significantly more effective in adapting policy behavior to human feedback. Building on these observations, we introduce **STEER** (**S**upervised **T**akeovers for **E**fficient **E**xploration in **R**einforcement Learning) - a hybrid framework that combines RL on policy experience with weighted IL on human interventions. Intuitively, the policy is quickly steered by recent human corrections to guide exploration (down weighting stale older corrections), and as the robot makes progress towards the task, it relies primarily on RL, making it robust to noise and human–learner divergence.

In summary (see Figure 1), this work addresses a fundamental need in sequential decision-making: learning effectively from human feedback with reliable evaluation before deployment. To this effect, we introduce a data-driven model of human interventions that captures how human behavior is influenced by agent's progress, enabling more reliable benchmarking for learning methods. Then, we propose a simple algorithm that uses BC regularized off-policy RL, using corrections to guide exploration rapidly rather than to prescribe the final objective. Across simulated manipulation tasks and human experiments, this framework improves sample efficiency and reduces human effort while remaining robust to noisy, divergent interventions.

## 2 RELATED WORK

**Learning from Human Demonstrations**    Behavioral cloning on offline datasets (Argall et al., 2009; Osa et al., 2018; Memmel et al., 2025) is widely-used to train robot policies, but it suffers from compounding errors during deployment due to data-distribution shifts (Ross et al., 2011). Complementary to this, RL enables training robust policies via task-rewards (Haarnoja et al., 2018a;b; Schulman et al., 2017), but is inefficient and uses undirected exploration to search for successful behavior in high-dimensional spaces. Recent works (Rajeswaran et al., 2018; Nair et al., 2021; Kostrikov et al., 2022; Yin et al., 2025; Ball et al., 2023) have leveraged offline datasets to initialize policies, value functions and replay buffers to warm start the RL process. Particularly, Lu et al. (2022) uses behavior cloning on offline datasets with off-policy (Haarnoja et al., 2018a) actor-critic updates to guide RL. While these methods effectively leverage human data to improve sample efficiency, collecting task demonstrations is hard and expensive. In contrast, we leverage easy-to-provide, online interventions (Michael et al., 2019) to bias the policy behavior during training. Finally, Fujimoto & Gu (2021) show that BC-regularized off-policy RL is a strong method, and in this paper we show the surprising effectiveness of our hybrid method with eligibility-style decay for *online interventions*, grounded in empirical human behavior.

**Interactive Learning**    Interactive learning methods (Ross et al., 2011; Michael et al., 2019; Jiang et al., 2024; Luo et al., 2024b; Xie et al., 2022; Ainsworth et al., 2019) address the drawbacks of offline behavior cloning by collecting additional feedback during deployment, theoretically reducing the compounding error problem from quadratic to linear regret with respect to the episode horizon (Ross et al., 2011). Correcting per-step actions is hard for robots, so, a class of methods (Michael et al., 2019; Mandlekar et al., 2020; Xie et al., 2022; Liu et al., 2023; Luo et al., 2024b; Spencer et al., 2022) allow humans to take over the robot control and override the policy behavior with corrective actions. Michael et al. (2019); Mandlekar et al. (2020); Liu et al. (2023) use this correction to directly supervise the policy via behavior cloning, while Luo et al. (2024b); Bajcsy et al. (2018); Korkmaz & Biyik (2025) build models of this human behavior to guide RL. Lindner et al. (2022); Spencer et al. (2022); Ainsworth et al. (2019) apply constraints over the policy and value functions based on human corrections. Li et al. (2022) derive reward directly from human takeovers to drive policy improvement, but still requires slow actor-critic updates. Additionally, reward-free HIL methods Peng et al. (2023; 2025) modify the critic with proxy targets, which make policy learning potentially Bellman-inconsistent and unstable under noisy/incorrect interventions. Overall, these methods make strong assumptions about the optimality of humans. Notably, Luo et al. (2024b) makes no such assumptions and adds interventions to the replay buffer to speed-up off-policy RL, achieving impressive real-world results. This provides further evidence that the intervention models used to guide the development of learning algorithms have been erroneous. In this work, we directly compare to Luo et al. (2024b), and find that their off-policy RL approach significantly slows learning from demonstrations relative to STEER .

**Modeling Human Behavior**    Learning from human interventions requires understanding the dynamics of human intervention for data-driven algorithm design and evaluation. Because real-world

training is costly (Yin et al., 2025; Torne et al., 2024), prior work has relied on approximate human models. TAMER (Knox & Stone, 2009) framed human feedback as a prediction model of scalar rewards over transitions. Bajcsy et al. (2018); Bıyık et al. (2022); Wilson et al. (2012) formulate interventions as Boltzmann-rational functions proportional to the value of states and actions. Korkmaz & Biyik (2025) proposes a probit model that intervenes when the human's nominal action is substantially better than the robot's expected value. Similarly, RLIF (Luo et al., 2024a) models interventions occur when the agent action falls below a threshold value of the optimal action. A shared assumption is that the model is Markovian and reactive to instantaneous sub-optimality. Our analysis of real intervention data contradicts this, indicating that such models are ill-suited for evaluating training methods. Further, cognitive theories suggest such decisions involve utility tradeoffs (Kahneman & Tversky, 1979), and recent caregiving models condition intervention on learner ability and task utility over time (Shachnai et al., 2025). Consistent with these insights, our proposed model takes into account the agent's progress over a short horizon to choose interventions. We ground our framework in real human intervention data, and these findings motivate our algorithmic framework.

## 3 PROBLEM STATEMENT

In this work, we build on the framework of interactive imitation learning (Michael et al., 2019; Luo et al., 2024a;b; Mandlekar et al., 2020; Liu et al., 2023), focusing on learning from *human interventions*. We consider an interactive control setting over an MDP $\mathcal{M} = (\mathcal{S}, \mathcal{A}, \mathcal{T}, \gamma, \rho_0)$, where $\mathcal{S}$ is the state space, $\mathcal{A}$ the action space, $\mathcal{T}$ the transition dynamics, $\gamma$ the discount factor, and $\rho_0$ the initial state distribution. The objective is to learn a policy $\pi_\theta(a \mid s)$ that maximizes expected discounted return, i.e., $\pi_\theta = \arg\max_\theta \mathbb{E}_{\pi_\theta}\left[\sum_{t \geq 0} \gamma^t r(s_t)\right]$. During deployment, a human observes the agent and may intervene. We model the human as a decision function $\mathcal{H} = (g, \pi_h)$ with two components: a *gating function* $g$ that decides *when* to intervene, and a *human policy* $\pi_h$ that decides *how* to intervene. To allow temporal context, both may depend on a recent history $\tau_{t-L:t} = (s_{t-L}, a_{t-L}, \ldots, s_t)$, so, $g : \mathcal{S}^{L+1} \times \mathcal{A}^L \to [0, 1]$ outputs the probability of intervening at time $t$, and $\pi_h : \mathcal{S}^{L+1} \times \mathcal{A}^L \to \mathcal{A}$ returns a distribution over actions. The resulting rollout policy is the mixture $\pi'(a \mid s_t, \tau_{t-L:t}) = g(\tau_{t-L:t})\,\pi_h(a \mid \tau_{t-L:t}) + (1 - g(\tau_{t-L:t}))\,\pi_\theta(a \mid s_t)$, which makes no optimality assumption about the human. Interventions can be variable-length (single actions or short segments). Thus, modeling human behavior requires simulating $g$ and $\pi_h$.

Prior baselines often assume access to $(\pi^*, Q^*)$, and set $\pi_h \equiv \pi^*$ (an assumption that we will show is invalid using real data demonstrating human suboptimality and noise, see Section 4), and define $g$ as a Markovian function of $Q$ and $\pi$. Concretely, Bajcsy et al. (2018) models $g(s_t, a) \propto \exp\{Q^*(s_t, a)\}$ as a Boltzmann-rational function, while Korkmaz & Biyik (2025) introduces a probit choice model based on $g(s_t, a) = \Phi\big(Q^*(s_t, a) - \mathbb{E}_{a' \sim \pi(\cdot \mid s_t)}[Q^*(s_t, a')] - c\big)$, and Luo et al. (2024a) models $g$ based on action sub-optimality i.e. $g(s_t, a_t) = \mathbb{1}\big[Q^*(s_t, a_t^*) - Q^*(s_t, a_t) > \tau\big]$ where $\tau, c$ are hyperparameters and $\Phi$ is the standard normal CDF. These formulations make strong assumptions about Markovian structure and $g$ being proportional to per-step sub-optimality, which is not grounded in real-world data and is misaligned with observed human behavior, rendering them suboptimal for both learning and evaluation. In the following sections, we (i) collect and analyze real human interventions to characterize when and how people intervene, (ii) propose a temporally grounded, progress-based model for $g$ validated against this data, and (iii) introduce an algorithm that addresses the key failure modes of prior approaches in learning from interventions.

## 4 HOW DO HUMANS INTERVENE FOR ROBOT POLICY LEARNING?

As discussed in Section 3, prior works (Knox & Stone, 2009; Luo et al., 2024a; Spencer et al., 2022; Korkmaz & Biyik, 2025) make flawed assumptions without grounding it in real-world comparisons. However, Luo et al. (2024b) relaxes any assumption to use correction transitions as off-policy data for RL (Ball et al., 2023). This method is surprisingly effective, significantly outperforming past methods. This provides further evidence that these assumptions are invalid and hence not beneficial in practice. This raises an important question: how and when do humans actually intervene? In order to address this, in the following sections (Section 4.1) we perform a comprehensive analysis of real human intervention data. We observe that human behavior is sub-optimal (Section 4.2) and non-Markovian (Section 4.4), which contradicts prior assumptions. Following this, in Section 4.4

we propose a better model for simulating human intervention that better fits offline data, enabling more reliable simulated evaluation.

## 4.1 COLLECTING REAL HUMAN INTERVENTION DATA

We run human-in-the-loop robot manipulation experiments to collect intervention data, measuring *when* people intervene and *how* they act during takeovers to evaluate and build better intervention models to inform our learning method. To isolate human behavior from hardware noise, we run studies on three simulated robot tasks (see Section 6) with four human participants. Similar to Luo et al. (2024b), we train a robot policy using off-policy RL and each user can take over from the learning policy at anytime and control the robot end-effector with a 3D space mouse. Humans observe and intervene until the policy converges to success. Further, we train a reference expert policy and value function using RLPD (Ball et al., 2023) on each simulated task to get unbiased experts $(\pi^*, V^*)$ that aid in our analysis.

## 4.2 HOW DO HUMANS INTERVENE?

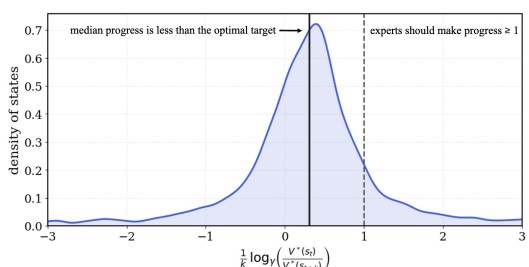

Generally, the intervention policy $\pi_h$ is assumed to be (near-)optimal (often identified with $\pi^*$) so that human corrections can be treated as demonstrations. This assumption underlies several methods discussed in Section 2. However, before adopting this for learning or evaluation, we conduct an empirical test on real intervention data. To assess task-agnostic optimality, we introduce a measure of value improvement over a trajectory segment $(s_t, a_t, s_{t+1}, \ldots a_{t+k-1}, s_{t+k})$. Under sparse rewards, an optimal $k$-step segment from $s_t$ to $s_{t+k}^*$ satisfies $V^*(s_t) = \gamma^k V^*(s_{t+k}^*)$, whereas

Figure 2: Distribution of progress over real intervention segments (higher is better). The median below the optimal suggests that many interventions are not near-optimal.

any other sequence of $k$ actions yields $V^*(s_t) \geq \gamma^k V^*(s_{t+k})$. This motivates the length-normalized progress score: $\text{Progress}(s_t \to s_{t+k}) = \frac{1}{k} \log_\gamma \left( \frac{V^*(s_t)}{V^*(s_{t+k})} \right)$, where larger values indicate greater improvement (higher is better). The metric compares an observed intervention segment against the optimal $k$-step baseline without assuming access to optimal actions.

We compute the progress score for human intervention on the dataset collected in Section 4.1. As observed in Fig. 2, the distribution of progress has a low median value ($\sim 0.3$) and mean value ($\sim -0.0127$) (as compared to the optimal target $\geq 1$), indicating that many human interventions generally fail to achieve near-optimal progress. Also, in Figure 4, we observed that the $V^*(s_t)$ drops along human corrections indicating that humans are not always optimal. This contradicts the common assumption $\pi_h \equiv \pi^*$, implying that to learn from humans: methods should be robust to non-expert, noisy corrections leading to our method in Section 5 that effectively combines BC and RL to learn from interventions.

## 4.3 HUMANS AND POLICIES DIVERGE DURING TRAINING

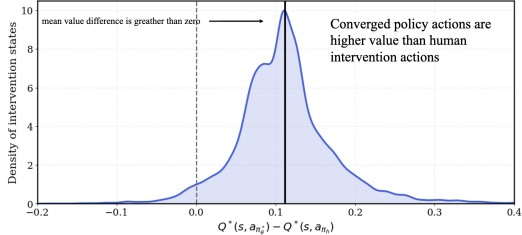

We further probe an additional source of misalignment between the learner policy and human beyond suboptimal progress. In Figure 3, we compare the value of actions from the converged policy $\pi_\theta^*$ and human corrected action under the converged critic i.e. $Q^*(s, a_{\pi_\theta^*}) - Q^*(s, a_{\pi_h})$ over all the states from training. We observe that distribution is concentrated above zero, suggesting that on convergence the learner favors a different higher valued action (or essentially, a different behavior mode) than the human. Notably, we observed that the task

Figure 3: Distribution of $Q^*(s, a_{\pi_\theta^*}) - Q^*(s, a_{\pi_h})$ at intervention states during training (larger values favor learned $\pi_\theta$). Most values $\geq 0$ imply converged policy actions are higher valued than human corrections, showing systematic divergence.

success during training is high ($\sim 76\%$), implying that the difference reflects a divergence in solution paths rather than execution failures.

Our hypothesis is that sparse-reward tasks have many optimal solutions; e.g., a cup-picking task can succeed via left- or right-side grasps. The human supervisor neither observes the full policy nor the exact underlying reward and thus, potentially provides conflicting behavior as a correction. Empirically, this manifests as policy actions having systematically higher value than humans under the converged critic. Imposing strict supervision in such settings can induce undesirable mode averaging for unimodal policy classes (Osa et al., 2018). Increasing policy expressivity (Chi et al., 2024) to capture multi-modality raises complexity.

These findings motivate a regularization on the supervision from intervention signals; specifically, they should accelerate early exploration and recovery, but not affect convergence behavior. In Section 5, we introduce a decay function that regularizes this supervision, shifting the bias from imitation towards independent trial and error learning as training progresses, decreasing any divergence issues caused by the human.

### 4.4 WHEN DO HUMANS INTERVENE?

In Section 3, we discuss that prior work typically posits that the when-to-intervene model $g$ is Markovian and reacts to instantaneous suboptimality. This implies that timing depends only on the current state (and possibly the current action), not on how the agent has been performing over time. Our empirical analysis contradicts this view. In Figure 4, we observe that interventions concentrate after short plateaus or drops in $V^*(s_t)$, indicating that it is a non-Markovian function that depends on progress over a horizon. In addition, we observe that humans tend to intervene when progress stagnates or drops, which motivates our following formulation. Also, Knox & Stone (2009) showed that human response for corrections is delayed relative to agent behavior. Together, they motivate the following claim: **intervention timing is non-Markovian and progress-sensitive**, and instantaneous suboptimality is insufficient.

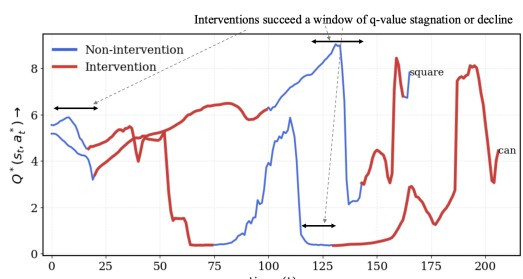

Figure 4: Interventions follow short $V^*$ plateaus or drops, indicating non-Markovian, progress-dependent functions, shown on trajectories from the can and square tasks.

**Proposed gating model:** Thus, we instantiate $g$ as a simple stagnation-based model that depends on the recent history $\tau_{t-k:t}$. Let $V^*(s_t) - V^*(s_{t-k})$ denote progress over horizon $k$. We define $g$ as:

$$g(\tau_{t-k:t}) \; = \; \Pr(\nu_t{=}1 \mid \tau_{t-k:t}) \; = \; \begin{cases} \alpha, & \text{if } V^*(s_t) - V^*(s_{t-k}) < \delta, \\ \beta, & \text{otherwise,} \end{cases},$$

where $k$ is the horizon and $\delta$ a progress threshold; $\alpha, \beta$ capture stochasticity in human decisions. Intuitively, the model intervenes when the agent's recent behavior fails to increase value (stagnation). Also, this formulation naturally covers failures in the robot behavior as $\delta > 0$ and failures would imply $V^*(s_t) - V^*(s_{t-k}) < 0$.

We evaluate this model against two baselines: Action Suboptimality (used in (Luo et al., 2024a; Korkmaz & Biyik, 2025)) as well as a **Random** model calibrated to the empirical intervention rate. Using human data from Section 4.1, we fit each model's parameters and report intervention prediction precision and recall via grid search (Details in Appendix B.2.3, and report the precision and recall across all methods.

| Method | Precision | Recall |
|---|---|---|
| Action-Suboptimality | 0.058 | 0.4848 |
| Random | 0.061 | 0.061 |
| **Ours** | **0.141** | **0.599** |

Table 1: Intervention prediction performance on held-out human data. The progress-sensitive, non-Markovian model better matches real timing.

**Our progress-based when-to-intervene model is a stronger evaluation model for simulation.** In Table 1, we observe that our model (evaluated on the held out task-data) outperforms the baselines in terms of recall and precision, indicating that

a model accounting for robot progress over a short horizon correlates better with human behavior. Our stagnation-based model attains higher agreement with real timing, consistent with the non-Markovian, progress-sensitive nature of human intervention. This shows that our model can be used as an effective method to simulate human behavior and test different intervention learning methods.

**Overall, our analysis informs the algorithm design to robustly use suboptimal, divergent human corrections and our better when-to-intervene model leads to stronger evaluation.**

## 5 How to use human interventions for robot policy learning?

Our analysis in Section 4 shows that interventions are sub-optimal and we cannot use them as demonstrations or reward surrogates. We therefore propose a method that adapts rapidly to noisy online interventions to guide behavior, while preserving the true unbiased objective. Following drawbacks of prior works (Section 4), we propose a method for fast adaptation to human interventions via off-policy actor critic loss on online policy experience and a weighted behavior cloning loss on human corrections.

Practically, we learn a policy $\pi_\theta$ and critic $Q_\psi$ using an online off-policy actor–critic algorithm (Haarnoja et al., 2018b;a; Ball et al., 2023). In our framework (see Algorithm 2), online roll-outs of $\pi_\theta$ are added to an experience buffer $D_\pi$. As a human supervisor (or simulated human model) intervenes, providing corrective actions, we store such transitions in a separate buffer $D_{\text{intervene}}$. Optionally, we can initialize $D_\pi$ and $D_{\text{intervene}}$ with a small amount of offline demos, similar to Ball et al. (2023) to warm-start the RL training. The critic $Q_\psi$ is updated on $D_\pi$ by minimizing the Bellman regression loss $\psi \leftarrow \arg\min_\psi \mathbb{E}_{(s,a,r,s')\sim D_\pi}\big[\big(Q_\psi(s,a) - (r + \gamma\,\mathbb{E}_{a'\sim\pi_\theta(\cdot|s')}[Q_\psi(s',a')])\big)^2\big]$. For the actor, the learning objective combines (i) standard RL policy improvement over $D_\pi$ that optimizes for task-success (ii) a maximum-likelihood objective on $D_{\text{intervene}}$ that biases exploration:

$$\theta \;\leftarrow\; \arg\max_\theta \; \underbrace{\mathbb{E}_{s\sim D_\pi,\,a\sim\pi_\theta(\cdot|s)}\big[Q_\psi(s,a)\big]}_{\text{RL policy-improvement on } D_\pi} \;+\; \underbrace{\lambda(i)}_{\text{time-varying weight}} \underbrace{\mathbb{E}_{(s,a)\sim D_{\text{intervene}}}\big[\log\pi_\theta(a\mid s)\big]}_{\text{maximum-likelihood alignment on interventions}}$$

Intuitively, standard actor-critic uses sampling of actions from stochastic policies for exploration (which is very inefficient in high dimensional spaces (Ladosz et al., 2022)). Our method additionally uses the supervision signal from interventions to update the policy directly and guide its exploration. Human interventions steer the agent towards promising regions of the environment, and help reduce failure or accelerate task completion, so they need not be optimal. Consequently, this also avoids the need for slow critic updates over the intervention segments to align the policy to human feedback. Thus, our method enables fast and robust adaptation to interventions.

**Importance of $\lambda(i)$** STEER makes explicit that interventions directly shape the policy via a fast, supervised update. To be effective under noisy feedback, the learning objective should preserve policy invariance to such shaping (Ng et al., 1999). We therefore regularize supervision with a weight $\lambda(i)$, so that intervention-driven updates accelerate early exploration but vanish asymptotically.

To this effect, for each intervention pair, $(s, a^{\text{human}})$, we store supervision weight that decays geometrically across actor updates, (similar to eligibility traces in tabular RL (Sutton, 1988; Sutton & Barto, 2018)) i.e. $\lambda(t+1|s, a^{\text{human}}) = (1 - \epsilon) * \lambda(t|s, a^{\text{human}})$, where $\epsilon$ is the decay parameter. As a result, as training proceeds, $\lambda(i) \to 0$, so at convergence (when humans stop intervening) the actor optimizes only the RL objective. Intuitively, as discussed in Section 4.3, human and learners can diverge over training, so this decay scheme naturally gives higher influence to recent interventions while diminishing stale corrections; reducing bias from noise and mode mismatch.

## 6 Experiments and Results

In our experiments, we aim to answer the following questions: (1) Does STEER achieve better task performance across both real and simulated humans? (2) Does STEER reduces human effort in terms of intervention effort while achieving similar performance? and (3) What roles does $\lambda(i)$ play in STEER ?

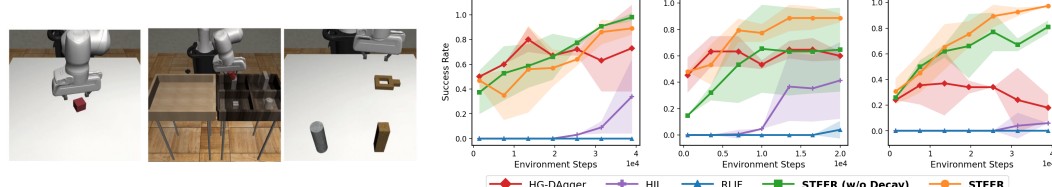

Figure 5: Left: Three robotic manipulation tasks which we use to collect intervention data and evaluate our methods. Right: Success rate vs environment steps across multiple tasks and baselines with simulated humans. STEER consistently outperforms all baselines.

**Tasks**   We evaluate our method across three simulated robotic manipulation tasks (**with both simulated and real humans**), adapted from standard benchmarks (RoboMimic (Mandlekar et al., 2020)): 1) **Lift:** Grasp and lift a cube to a target height. 2) **Can:** Pick up a cylindrical can and place it into a designated bin. 3) **Square:** Insert a square peg into a square hole, testing fine-grained manipulation and alignment.

**Baselines**   In our experiments, we compare to multiple competitive baseline methods across real and simulated human interventions: 1) **HIL** (Ball et al., 2023): A hybrid method combining off-policy RL with online human-in-the-loop data in the replay buffer. 2) **HG-DAgger** (Michael et al., 2019): An extension of DAgger (Ross et al., 2011) where human interventions are used as additional demonstrations for behavior cloning. 3) **RLIF** (Luo et al., 2024a): A method that updates the reward function in HIL, treating human interventions as negative reward signals at takeover states. 4) **STEER** and **STEER (w/o decay)** : Our method introduced in Section 5, along with an ablation with an unweighted supervised loss.

**Training and Evaluation Details**   We use a jax-based implementation of HIL Luo et al. (2024b) as our base off-policy RL algorithm. In Section 4.1 and Appendix B.2.1 we carefully outline our details for collecting real-world data and experiments. For experiments with simulated humans, we use our proposed intervention model from Section 4.4 and a sub-optimal RL policy to provide corrections to the learner and evaluate our method across three seeds across the three tasks. For real world experiments, we run all methods with two humans across two tasks. Since we want to investigate methods that learn from real humans, where data is expensive and iteration speed is critical, we investigate all methods under limited environment interaction budgets with 40k total steps in simulation and 15k steps in the real world. We include all the training details (including hyperparameters) in the Appendix B.2.

### 6.1 STEER achieves better task performance with both sim and real humans

As discussed above RLIF (Luo et al., 2024a) and HG-DAgger (Michael et al., 2019) require near-optimal human interventions, while HIL Luo et al. (2024b) purely uses inefficient actor-critic updates to learn from the interventions. In contrast, STEER deploys a hybrid approach to use interventions for guiding exploration in addition to off-policy RL. So, we see in Figure 5 that with simulated humans, STEER consistently outperforms all the baselines in terms of success rate, demonstrating the effectiveness of the intervention supervision in STEER to guide exploration during training. Michael et al. (2019) has non-trivial performance but falls short because it falsely assumes optimality in corrections. Finally, HIL is very inefficient, while RLIF has negligible performance within the limited budget of ∼40k environment interactions.

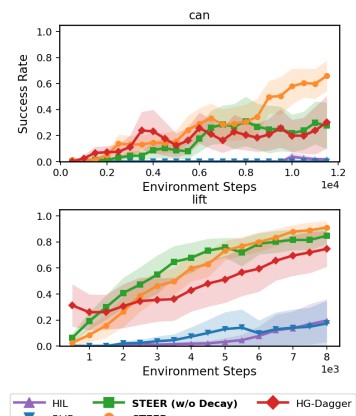

Figure 6: Comparing success rate vs environments with real humans intervening in the simulated tasks.

In Section 4.2, we observe that our intervention model for simulated experiments fits an offline dataset better. So, next we investigate if the conclusions about the algorithms derived from this setup translate to the real-world. In Figure 6, we observe that STEER significantly outperforms the baselines, and partic-

ularly, in the can task (with a longer horizon) the baselines completely fail while STEER converges to a high success rate across both human participants. As our simulated when-to-intervene model is closely aligned to real-world data, we also observe a strong correlation in the performance of different methods across the real and simulated settings, further highlighting the benefits of building better evaluation models.

In the simulated setting, the human correction behavior is a noisy model to emulate real-world humans that are noisy and irrational, and we observe that STEER is significantly robust to the sub-optimality in corrective actions. In comparison, the baselines that assume strong optimality in human data, perform poorly, indicating that STEER is a better approach to handle noisy humans.

## 6.2 STEER SIGNIFICANTLY DECREASES HUMAN EFFORT TO ACHIEVE THE SAME PERFORMANCE OVER OTHER METHODS.

Luo et al. (2024b) simply uses the intervention segments as off-policy data to update the replay buffer during off-policy RL. In order to use this data, the only way is to propagate the rewards achieved in these segments to the value function via Bellman backups (see Algorithm 1). And then the policy extraction step aligns the robot to execute these high-value functions in the human corrections.

In Figure 7, we observe that STEER leveraging supervised updates outperforms the RL baselines significantly in terms of human effort i.e. converges to a high success rate with ∼2x less human interventions. As STEER uses maximum-likelihood supervision to directly align the policy with the human corrected transitions (with a decaying weight to avoid optimality assumptions, unlike Michael et al. (2019)), it allows the policy to learn even faster with the online experience. For additional plots, see Appendix A.5.

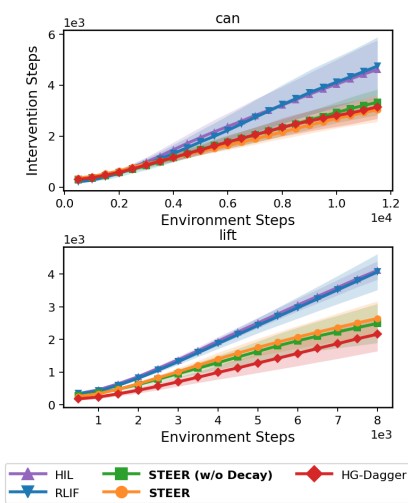

## 6.3 DOES $\lambda(i)$ ENABLE STEER TO BE ROBUST TO SUB-OPTIMALITY AND DIVERGENCE IN HUMAN INTERVENTIONS?

In Section 4.3 and 4.2, we observe with real-world data that as training progresses there is a divergence between the learner and the user. STEER accounts for this human divergence and irrationality via the decay parameter $\lambda(i)$. As a result, in Figure 6, when we train robots with real humans we observe that STEER outperforms the ablation

Figure 7: Comparing the number of interventions from real-humans across environment steps.

which does not incorporate this regularization during training. Our hypothesis is that this decay naturally weights the current interventions higher, while down-weighting the past interventions. As the training progresses, and intervention rates are lower, the policy update is completely dependent on the actor-critic updates, removing any bias introduced by the human intervener.

## 7 REAL-ROBOT EXPERIMENTS

We validate STEER on a real Franka Emika Panda (see Figure 8) performing a *pen-in-bowl* manipulation task, following the general protocol of Luo et al. (2024b). We use a wrist camera and a third camera for observations, and end-effector action space for controlling the robot. At the beginning of each episode, the pen's pose is randomized within a bounded workspace region while the bowl remains fixed. All methods are warm-started with 20 human demonstrations. During online learning, a supervisor provides takeovers via a 3D SpaceMouse. Each method interacts with the environment for 10,000 steps with a synchronized update-to-data ratio of 10. For evaluation, we execute 20 episodes at 5k and 10k environment steps and report success rates, as well as the number of human intervention steps.

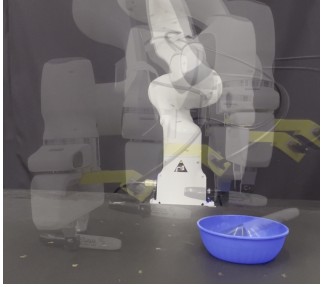

Figure 8: Real-robot experiment (pen-in-bowl) task, based on Luo et al. (2024b).

Table 2: Success rates on pen-in-bowl after 5k and 10k environment interaction steps (20 evaluation episodes per setting).

| Method | Success rate at 5k env steps | Success rate at 10k env steps |
|---|---|---|
| HIL | 0.15 (3 / 20) | 0.55 (11 / 20) |
| HG-Dagger | 0.25 (5 / 20) | 0.65 (13 / 20) |
| STEER (w/o decay) | 0.35 (7 / 20) | **0.90(18/20)** |
| STEER | **0.70(14/20)** | 0.85 (17 / 20) |

In Table 2, we observe that STEER achieves almost 50% higher success rate than HG-Dagger Michael et al. (2019). Figure 7 reports intervention steps over the environment interactions. STEER achieves high success with fewer intervention steps than all baselines, converging with roughly 2× less human effort. This efficiency stems from the hybrid update: the fast-supervised update enables rapid adaptation to human corrections, while the RL update allows the policy to leverage suboptimal demonstrations. Consistent with our analysis on simulated and human-in-the-loop settings, STEER remains robust to suboptimal and noisy corrections. These observations show that STEER is an effective and practical algorithm for online robot learning from corrections.

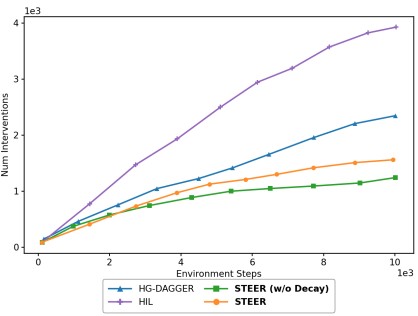

Figure 9: Real-robot experiment (pen-in-bowl): intervention steps vs. environment steps. STEER attains high success with significantly less human interventions than baselines.

## 8 DISCUSSION

This work performs a detailed study on the problem of learning from human interventions. We show that many of the optimality assumptions made in prior work about the nature of human interventions do not match data from actual human users. We conduct a detailed analysis of when and how human users intervene, showing that they focus much more on a notion of progress and stagnation than optimality. We use these findings to (1) instantiate a better correlated simulated human model for future researchers to develop methods against, (2) instantiate a new method for learning from human interventions that guides exploration rather than assumes optimality. While these findings are promising, many avenues for future work remain. We plan to extend this work to study humans intervening on real robots at scale. We also need to understand the plurality of intervention types across many different human interveners. And finally, we propose a simple naive algorithm for incorporating interventions; a more sophisticated exploration algorithm incorporating targeted optimism would be promising to explore in future work.

## 9 REPRODUCIBILITY STATEMENT

To ensure reproducibility, we will release our code (which is built on-top of public jaxRL codebase), our dataset collected across multiple users and our intervention models to guide future algorithms. We outline the complete training algorithm, hyperparameters, environments, reward functions and dataset details in Sections 4.1, 6 and Appendix B.2.5, B.2.1.

## 10 USE OF LARGE LANGUAGE MODELS (LLMS)

We used LLMs for assistance to write code required to produce some of the experiments in this paper. Additionally, we used LLMs to format figures and language corrections in the LaTeX source.

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

# A APPENDIX

## A.1 ADDITIONAL BASELINES

We include a comparison of STEER to additional baselines:

1. HIL Luo et al. (2024b): A hybrid method combining off- policy RL with online human-in-the-loop data in the replay buffer

2. RLIF Luo et al. (2024a): A method that updates the reward function in HIL, treating human interventions as negative reward signals at takeover states.

3. HG-DAgger (Michael et al., 2019): An extension of DAgger (Ross et al., 2011) where human interventions are used as additional demonstrations for behavior cloning.

4. PVP Peng et al. (2025): An off-policy HIL method that augments Bellman error, with additional binary supervision targets.

5. SIRIUS and IWR Liu et al. (2023); Mandlekar et al. (2020): These are HIL-methods that do only maximum likelihood supervision updates to the policy.

In Figure 10, we observe that STEER is the best-performing policy. A major benefit of STEER over other baselines is that it makes minimal assumptions about the quality and pattern of human interventions and, as a result, can leverage a wide range of interventions with different levels of optimality. Peng et al. (2025); Liu et al. (2023); Mandlekar et al. (2020) assume that humans intervene optimally, as a result, underperform when the corrections are noisy. Meanwhile, STEER is robust to the suboptimal corrections and converges to a higher success rate much more sample efficiently (both environment steps and human intervention effort) than the baselines.

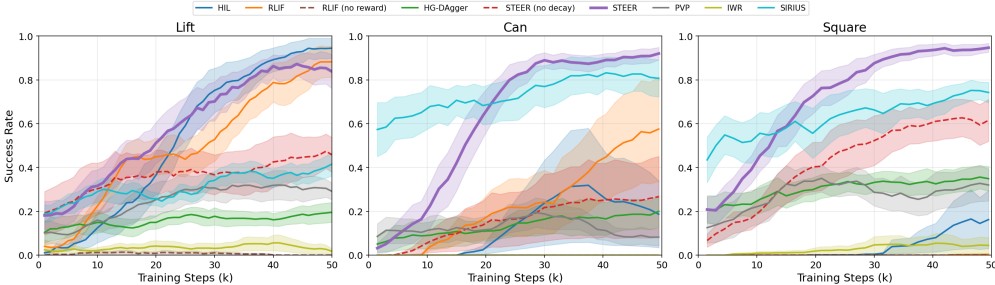

Figure 10: Additional baselines across three robomimic tasks in simulation. STEER consistently outperforms all baselines.

## A.2 HUMAN INTERVENTION MODELS

We run all the intervention learning algorithms across multiple human intervention models.

1. Random Intervention: a random model calibrated to the empirical intervention rate from the data in Section 4.

2. Action-Suboptimality based on Q-value: model based on Luo et al. (2024a).

3. Action-Difference: a model where the human intervention is simulated proportionally to $\|a^\pi - a^*\|_2$.

4. Our introduced gating model from Section 4.4

   In Figure 11, we observe that variants of STEER robustly have a high success rate, efficiently across different intervention styles. This further demonstrates that STEER makes no assumption about the corrective segments, and is a scalable and practical choice.

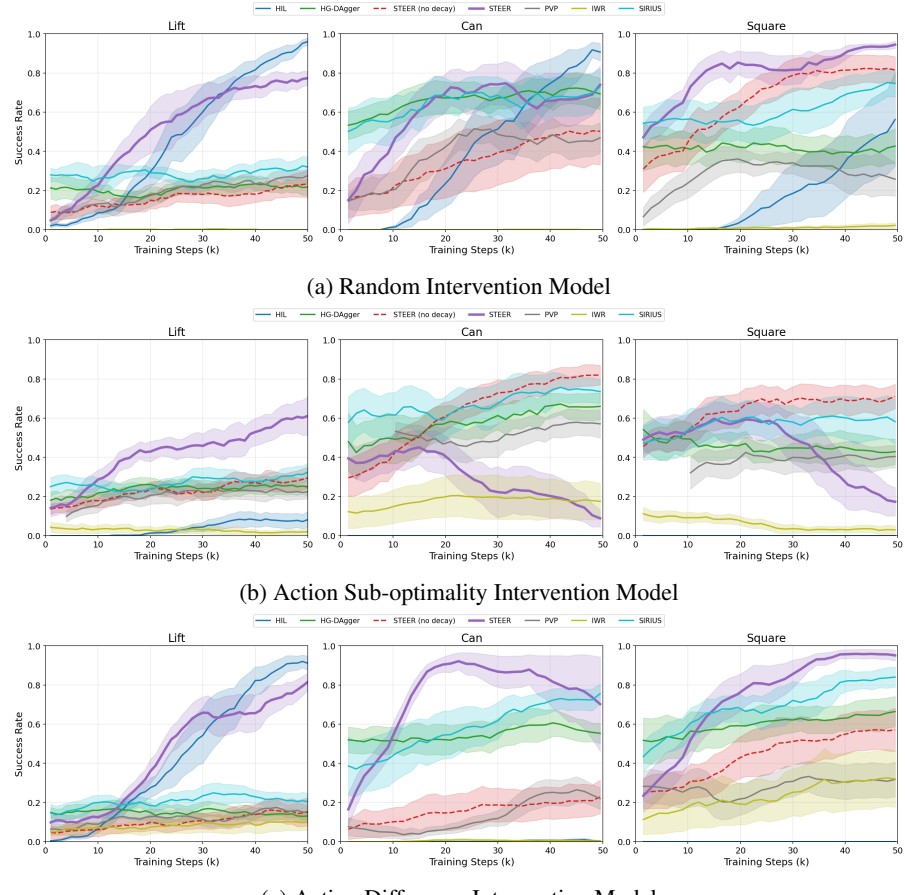

(a) Random Intervention Model

(b) Action Sub-optimality Intervention Model

(c) Action Difference Intervention Model

Figure 11: Right: Success rate vs environment steps across multiple tasks and baselines with simulated humans via different models. STEER consistently outperforms all baselines.

## A.3 CORRELATION BETWEEN SIMULATED HUMAN MODEL AND REAL HUMAN INTERVENTIONS

In Table 1, we compare the precision and recall of different human models. The main objective for building better human intervention models is: more reliable benchmarking of intervention learning algorithms. Following Li et al. (2024) that generates realistic simulators for evaluating real-world robot policies with higher success correlations between simulation and real-world evaluation, i.e., policies tested in simulation and real rank similarly – we perform a similar analysis across the rankings of the models across simulated human interventions and real human interventions across the two robomimic tasks.

| Method | Real Human Interventions | Stagnation | Q-Value | Action-Diff | Random |
|---|---|---|---|---|---|
| STEER | 1 | 1 | 3 | 1 | 1 |
| STEER (no decay) | 2 | 2 | 1 | 3 | 3 |
| HG-DAGGER | 3 | 3 | 2 | 2 | 4 |
| HIL | 4 | 4 | 4 | 4 | 2 |

Table 3: Rankings of the four algorithms under human evaluation and four simulated modalities.

To quantify how well each simulation signal predicts real-robot performance, we compute the Spearman rank correlation between the real-robot ranking and the ranking induced by each intervention model (see Table 3). In Figure 12, the stagnation-based gating model achieves the highest cor-

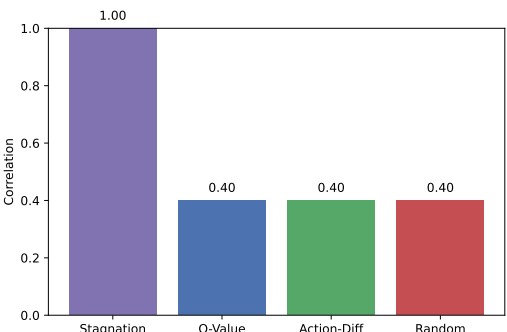

Figure 12: Correlation of ranking different algorithms across simulation and real interventions for different models of simulating human interventions.

relation with the real-robot ordering, indicating that it is the most reliable proxy for real-world performance, while Q-value, action-difference, and random rankings correlate substantially less.

## A.4  TRAINING WITH PERFECT EXPERTS IN SIMULATION

To evaluate the benefits of STEER with sub-optimal interventions, we run an ablation of using perfect experts in simulation. In our experiments, human sub-optimality is mirrored via sub-optimal RL checkpoints. In Figure 13, we compare the performance of all methods with perfect experts (RL-trained experts that are easy to imitate) and observe that most baselines achieve significant improvements in performance with close to optimal interventions, further emphasizing that STEER does not need the interventions to be optimal and outperforms the baselines under practical, noisy conditions.

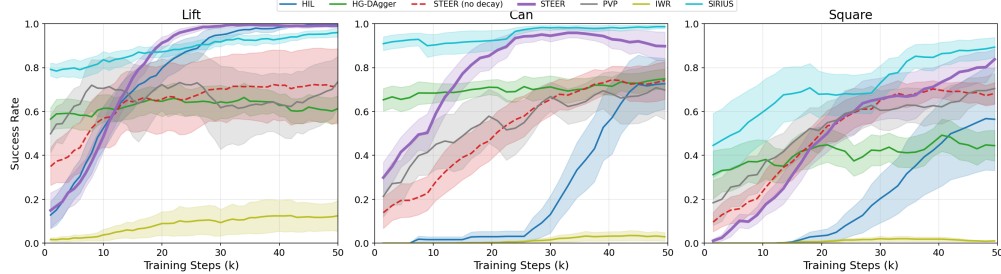

Figure 13: Comparing all algorithms with a perfect expert policy acting as the intervention policy in simulation.

## A.5  HOW DOES TASK PERFORMANCE CHANGE WITH MORE INTERVENTIONS ACROSS DIFFERENT METHODS?

In Figure 14, we observe that STEER uses significantly less human interventions to converge to a successful policy, while the RL baselines are worse (sometimes with zero performance within the same effort). HG-DAgger gets worse with more interventions because of noise in simulated human actions, showing the benefit of our method in handling noise in human actions which is confirmed, as discussed and verified in Section 4.

In Figure 15, we also plot the ratio between the expert q-value and the intervention action value as training progresses. We observe that human actions are sub-optimal i.e. have lower value compared to the optimal action at a fixed rate over the entire training run, and are not affected by time.

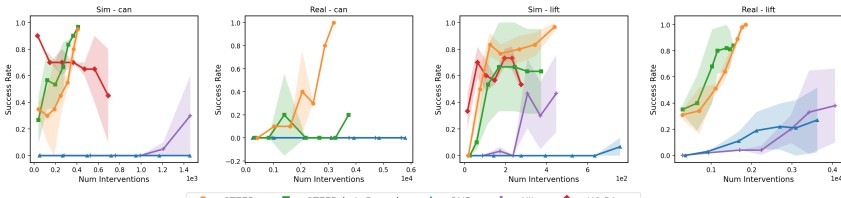

Figure 14: Comparing task success against number of real and simulated human interventions

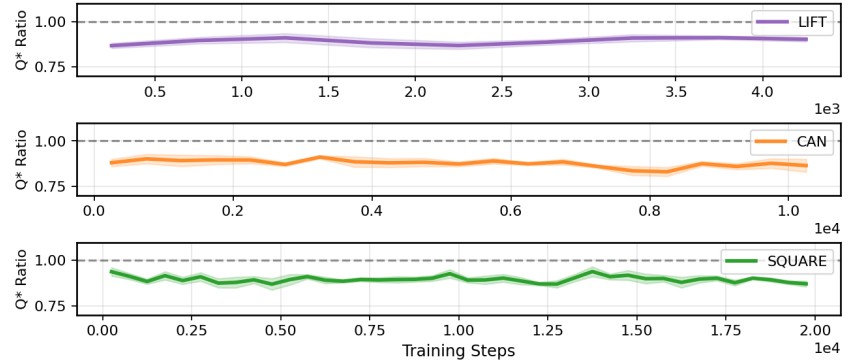

Figure 15: $\frac{Q^*(s, a_{\text{human}})}{Q^*(s, a^*)}$ as training progresses across the three tasks. Human sub-optimality is roughly similar across all training runs.

## A.6 ANALYSING INTERVENTION EFFORT ACROSS REAL AND SIMULATED HUMAN EXPERIMENTS

In Figure 16, we visualize the intervention rate of the either the simulated or the real human supervising the policy across different environments and algorithms. STEER aligns rapidly with human correction i.e. the supervised adapts bias the policy towards the behavior the human considers better early on the training leading to a drop in the intervention rate. But, this biased exploration enables the policy to collect useful and successful experience quickly, which is leveraged by the RL update later. As a result, STEER reach high success rates much faster than baselines. Further, across simulation and real we note qualitatively, that the intervention rate trends are correlated showing a stronger promise towards our introduced stagnation model being a better setting for evaluation.

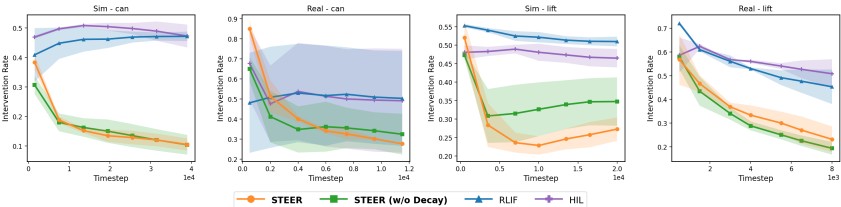

Figure 16: Comparing intervention rate for real and simulated human agaubst intervention steps

## A.7 INTERVENTIONS USING VR CONTROLLER

To evaluate the effect of different intervention interfaces, we also incorporate a virtual-reality–based teleoperation controller (Oculus). The human supervisor uses the controller's trigger to override the robot policy and provide corrective actions. As shown in Figure 17, the performance trends of STEER are similar across the two intervention devices, indicating that our insights and approach generalize across intervention devices and control modalities.

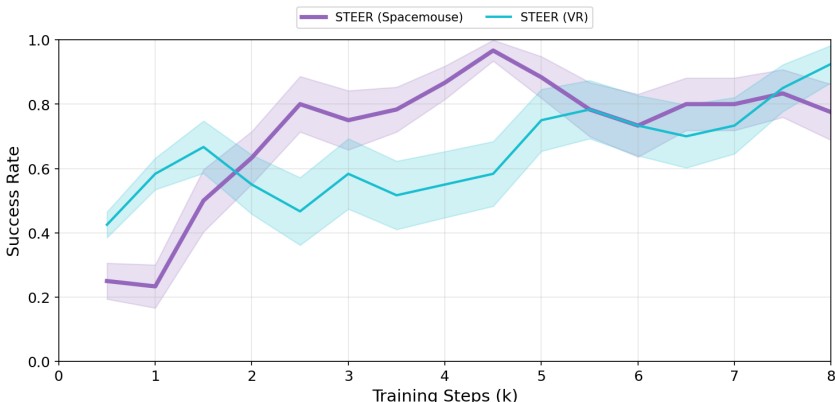

Figure 17: Comparing STEER with VR interventions and SpaceMouse on the Lift task. STEER achieves similar success rates with similar intervention effort.

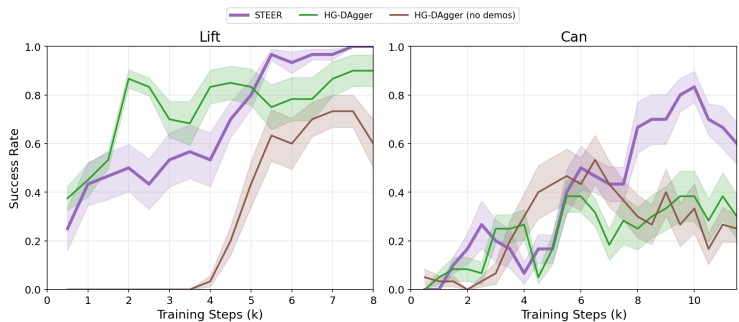

Figure 18: Comparing HG-Dagger, HG-Dagger (no offline data) to STEER for real human intervention on sim tasks

### A.8 ABLATION AGAINST TASK REWARD AND HUMAN SUPERVISION

We include experiments across the simulated can and lift tasks, with real human interventions on one additional baseline: HG-DAgger (no offline data): a variant of Michael et al. (2019), where we do not include the initial offline demos during training. In Figure 18, we observe that our method outperforms the baselines, while HG-DAgger with demos also outperforms the baselines, showing the benefit of effectively leveraging human interventions as well as offline demonstrations, during online training to accelerate policy convergence.

In Figure 19, we observe that even with sub-optimal interventions, a task-specific sparse reward enables RLIF Luo et al. (2024b) to perform better while the no-reward variant achieves negligible performance; however, it still makes unrealistic assumptions about the corrections, and STEER outperforms it across the two settings.

## B IMPLEMENTATION DETAILS

### B.1 TASK DETAILS

We evaluate our approach on three goal-conditioned robotic manipulation tasks. Each task is adapted from the RoboMimic benchmark suite (Mandlekar et al., 2020) and is posed with sparse, success-based rewards.

**Lift.** The agent must grasp a cube and elevate it to a target height above the table. The object observation is a 10-dimensional vector comprising the cube's absolute position and orientation, together with the cube's position relative to the end effector. The sparse reward is defined as $r = 1$ if the cube's height exceeds the target threshold.

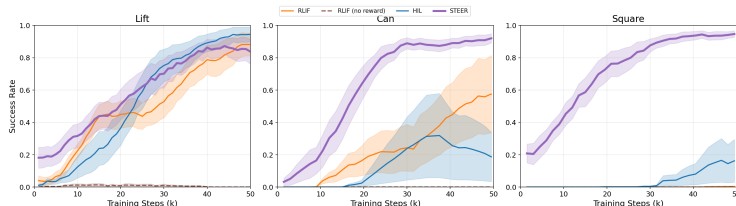

Figure 19: Comparison of RLIF with no task reward and with task reward against STEER under optimal interventions.

**Can.** The agent must pick up a cylindrical can and place it into a targeted bin region. The object observation is a 14-dimensional vector comprising the can's absolute position and orientation, as well as the can's position and orientation relative to the end effector. The task is successful if the can is within the target region.

**Square.** The agent must insert a square nut onto a square peg, requiring precise in-hand alignment prior to insertion. The object observation is the nut's absolute position, orientation, with the nut's relative position and orientation. At the beginning of each episode, the nut pose is randomized on the table. The objective is to align the nut with the peg within tolerance.

These tasks jointly assess different robot skills under sparse feedback, thereby providing a strong benchmark for intervention-based learning and for measuring improvements in sample efficiency and human effort.

### B.2 TRAINING DETAILS

#### B.2.1 REAL-WORLD SETUP

For experiments with real human interventions, we recruited researchers to intervene on robot policies using a 3Dconnexion SpaceMouse for teleoperation . Prior to data collection, participants underwent a familiarization phase that included: (1) collecting demonstration trajectories to understand the task dynamics and control interface, and (2) practice sessions intervening on pretrained policy checkpoints at various stages of learning to calibrate their intervention strategy.

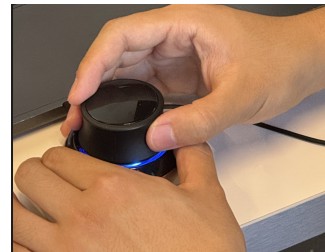

Figure 20: Researcher using a 3Dconnexion SpaceMouse for interventions

Our setup with real human interventions has two parts: data collection for intervention analysis and algorithm comparison.

First, we have participants intervene on policies trained using HIL (Luo et al., 2024b) as the base learning algorithm. To accelerate convergence, we initialized training with 25 demonstration trajectories, fixed the initial positions of both the robot end-effector and goal objects across episodes, and restricted the action space to only end-effector position deltas and gripper commands. This results in a dataset of human intervention behaviors across different stages of policy learning. We then analyze this data to create more accurate simulated interventions.

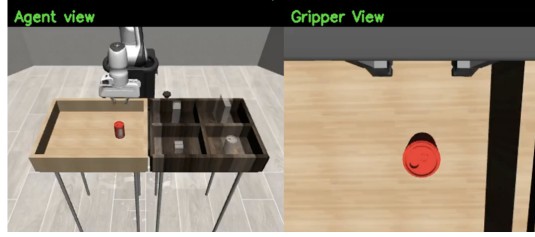

Figure 21: Human observations for intervening on a simulated robot task

Our second stage, algorithm comparison, has participants compare different learning algorithms on the lift/can task. For this stage, we randomize the initial position of the robot, and have the participant intervene for 8,000 steps for lift and 12,000 steps for can.

### B.2.2 SIMULATED HUMAN INTERVENER DETAILS

We intervened with a RLPD checkpoint achieving 50-70% success on the target task, and sample intervention lengths from the bottom 75% of intervention lengths.

### B.2.3 FITTING INTERVENTION MODEL PARAMETERS

To fit our progress-based intervention model to the collected human data, we performed a grid search over the window size $k$ timesteps and the progress threshold $\delta$, optimizing for the highest F1 score on intervention prediction. We developed a unified intervention model across all tasks rather than task-specific models, prioritizing the capture of fundamental intervention behaviors (e.g., reaction time, progress perception) over task-specific patterns to enhance generalization.

### B.2.4 REAL WORLD TRAINING

To evaluate our method with real human interventions, we conducted experiments where human participants intervened on learning policies in real-time. We tested four algorithms: HIL (Luo et al., 2024b), RLIF (Luo et al., 2024a), and our method STEER (with and without decay) across two manipulation tasks (Lift and Can) with two human participants.

We fixed initial goal positions and restricted the action space to 4DOF control (3D position deltas + gripper) to reduce complexity and improve learning speed. Participants used a 3Dconnexion SpaceMouse to provide interventions when they observed the robot making errors or failing to make progress toward the task goal.

We logged all interventions, policy rollouts, and task successes to analyze both final performance and intervention efficiency. To ensure consistency across experiments, we used the same hyperparameters for each algorithm as in our simulated experiments, with the only difference being the intervention source (human vs. simulated model).

### B.2.5 HYPERPARAMETERS

We outline all hyperparameters used in our experiments in Table 4.

Table 4: Hyperparameters for STEER and baselines. We use the same parameters across all experiments and report the best result on 3 seeds.

| Hyperparameter | Value |
|---|---|
| *Core RL Parameters* | |
| Architecture | MLP with Gaussian Head |
| Hidden layers | 3 layers of width 256 |
| Optimizer | Adam |
| Learning rate | 3e-4 |
| Discount ($\gamma$) | 0.99 |
| Soft update ($\tau$) | 0.005 |
| UTD Ratio | 5 |
| Batch size | 256 |
| Replay buffer size | 1e6 |
| *STEER-Specific Parameters* | |
| BC weight (initial $\lambda_0$) | 1.0 |
| BC decay rate ($\epsilon$) | 5e-4 (lift), 1e-4 (can) |
| BC decay type | Per-timestep exponential |
| *Intervention Model Parameters* | |
| Stagnation window ($k$) | 9 |
| Stagnation threshold ($\delta$) | -0.158 |
| True positive rate ($\alpha$) | 0.599 |

# C  ALGORITHM

---

**Algorithm 1** Human in the Loop Learning: HIL (Luo et al., 2024b)

---

**Require:** $\pi_\theta, Q_\psi, \pi^{\text{human}}, D_\pi$
1: **for** trial $i = 1$ to $N$ **do**
2:     **for** timestep $t = 1$ to $T$ **do**
3:         **if** $\pi^{\text{human}}$ intervenes at $t$ **then**
4:             append $(s_t, a_t^{\text{human}}, r_t, s_{t+1})$ to $D_\pi$
5:         **else**
6:             append $(s_t, a_t, r_t, s_{t+1})$ to $D_\pi$
7:         **end if**
8:     **end for**
9:     $\psi \leftarrow \arg\min_\psi \mathbb{E}_{(s,a,r,s')\sim D_\pi}\left[\left(Q_\psi(s,a) - (r + \gamma Q_\psi(s', \pi_\theta(s')))\right)^2\right]$
10:    $\theta \leftarrow \arg\max_\theta \mathbb{E}_{s\sim D_\pi, a\sim \pi_\theta}\left[Q_\psi(s,a)\right]$
11: **end for**

---

---

**Algorithm 2** STEER: Supervised Takeovers for Efficient Exploration in RL

---

**Require:** $\pi_\theta, Q_\psi, \pi^{\text{human}}, D_\pi, D_{\text{intervene}}, \lambda(.)$
1: **for** trial $i = 1$ to $N$ **do**
2:     **for** timestep $t = 1$ to $T$ **do**
3:         **if** $\pi^{\text{human}}$ intervenes at $t$ **then**  $\triangleleft$ or a simulated $(g, \pi_h)$ behavior model
4:             append $(s_t, a_t^{\text{human}})$ to $D_{\text{intervene}}$
5:         **else**
6:             append $(s_t, a_t, r_t, s_{t+1})$ to $D_\pi$
7:         **end if**
8:     **end for**
9:     $\psi \leftarrow \arg\min_\psi \mathbb{E}_{(s,a,r,s')\sim D_\pi}\left[\left(Q_\psi(s,a) - (r + \gamma Q_\psi(s', \pi_\theta(s')))\right)^2\right]$
10:    $\theta \leftarrow \arg\max_\theta \mathbb{E}_{s\sim D_\pi, a\sim \pi_\theta}\left[Q_\psi(s,a)\right] - \mathbb{E}_{(s,a)\sim D_{\text{intervene}}}[\lambda(i|s_t, a_t) * \log \pi_\theta(a \mid s)]$
11:                                             $\triangleleft$ weighted BC loss on interventions
12:    $\lambda(i+1|s_t, a_t) = (1 - \epsilon) * \lambda(i|s_t, a_t) \;\; \forall (s_t, a_t) \in D_{\text{intervene}}$
13: **end for**

---

