# OpenReview forum: "Towards Understanding Learning from Human Interventions"
_ICLR.cc/2026/Conference — ICLR 2026 Conference Withdrawn Submission_

### Official Review · Reviewer_hTA7 · 2025-10-25

**Soundness:** 3
**Presentation:** 3
**Contribution:** 2
**Rating:** 4
**Confidence:** 4

**Summary:**

The paper suggests that in intervention learning, human interventions are non-Markovian and depend on recent progress over the past few steps. The paper proposes STEER algorithm, which combines the Bellman regression loss with a behavior cloning loss on expert interventions. On both simulated and real human interventions, STEER outperforms the baselines across three robotic manipulation tasks.

**Strengths:**

1. Figures 2, 3, and 4 intuitively illustrate the human takeover mechanism, showing that human interventions are non-Markovian.
2. The paper conducts experiments on three manipulation tasks in Robosuite simulators, where STEER consistently outperforms the baselines.
3. STEER is easy to implement and has high compuational efficiency.

**Weaknesses:**

1. **The connection between the motivation and STEER algorithm is unclear**. The motivation in Fig 2,3,4 highlights the non-Markovian nature of the human takeover mechanism, but the STEER algorithm itself simply combines two losses from two separate buffers. It's unclear how STEER's design (line 341, page 7) is related to the proposed intervention mechanism (line 301, page 6). I don't see why STEER should be better than the baselines at handling non-Markovian interventions.
2. For the **simulated human experiments**, the paper **only uses one gating model** based on recent task progress (line 301) with a sub-optimal RL expert. It’s possible that under different gating models (e.g., using the action difference between the agent action and the RL expert action for intervention), baselines like HG-DAgger could outperform STEER.
3. STEER **relies on the task reward** for training, whereas baselines such as HG-DAgger and RLIF do not require it.
4. It's unclear **whether the proposed gating model can help the agent succeed during training**.  If the interventions are poor and fail to help the policy reach the goal, STEER may gain an unfair advantage over baselines because STEER can access the task reward and has a vanishing coefficient $\lambda(i)$. In order to justify the gating model, I would like to see the success rate of the behavior policy during training (with proper expert interventions, I expect that this training success rate approaches the expert's success rate).

**Questions:**

1. What is the reward model used to compute Q* and V* in Figures 2, 3, and 4? Is the reward very sparse?
2. Figures 2, 3, and 4 claim that humans are suboptimal. Could this result from the difference between the task reward and the human's intrinsic reward?
3. In Fig. 5 (Task Square) and Fig. 8 (Sim Can), the performance of HG-DAgger decreases as the number of human interventions increases. Could this result from an improper gating model, where the simulated human only provides data in non-critical states? As the agent's performance improves, how does the behavior of the gating model change?

---

> ### Author Response · Authors · 2025-11-28
>
> We would like to thank the reviewer for their careful review of the paper and for discussing the ease and scalability of the introduced method. We answer all concerns raised below:
>
> - **The connection between the motivation and the STEER algorithm is unclear.**
>
> Refer to pt B. of the common response.
>
> - **For the simulated human experiments, the paper only uses one gating model based on the recent task*..*
>
> Refer to pt C. in the common response.
>
> - **STEER relies on the task reward for training, whereas baselines such as HG-DAgger and RLIF do not require it.**
>
> We follow prior real-world RL works (HIL-SERL[1], PLD[2], etc.) in assuming access to a sparse task reward, which is both practical to obtain and critical for learning. Because human interventions are sub-optimal (Figures 2–4), task rewards provide the necessary signal for the policy to improve beyond human data; without them, the problem is underspecified, and optimal behavior cannot be recovered. In Figure 19, we show that with sub-optimal interventions, even RLIF benefits from a task-reward and is able to achieve non-trivial success rates, while the reward-free RLIF baseline fails.
>
> - **It's unclear whether the proposed gating model can help the agent succeed during training**
>
> The gating model controls when the human intervenes in the simulated gating model, and the policy controlling the intervention actions is a sub-optimal RL checkpoint with a non-trivial (\~50-80%) success rate. We observe that all gating methods result in successful and unsuccessful trajectories, with performance matching real human performance (\~80% for interventions on a real robot).
>
> - **What is the reward model used to compute Q\* and V\* in Figures 2, 3, and 4? Is the reward very sparse?**
>
> We use a sparse reward that gives 0 or 1 at the end of a trajectory based on it being successfully completing the task or not. We would make sure to update the manuscript with these details further.
>
> - **Figures 2, 3, and 4 claim that humans are suboptimal. Could this result from the difference between the task reward and the human's intrinsic reward?**
>
> This is a very interesting question. Past work [3] has shown that human interventions are different because of partial observability. This could also be attributed to differences in the reward functions or the system being underactuated. Overall, this is an interesting direction of work which we aim to study in the future.
>
> - **In Fig 5 (Task Square) and Fig. 8 (Sim Can), the performance of HG-DAgger decreases as the number of human interventions increases.**
>
> Our analysis of human interventions (Figures 2–4) shows that corrective segments are sub-optimal. As the number of interventions increases, HG-Dagger increasingly imitates this noisy data, with no mechanism to recover from suboptimal demonstrations. To validate this, Figure 13 uses perfect RL-trained experts—easy to mimic—and HG-Dagger’s performance improves substantially. However, such experts are impractical in real settings, where STEER consistently outperforms HG-Dagger (Table 3 and Figure 10).
>
>
> [1.] Precise and Dexterous Robotic Manipulation via Human-in-the-Loop Reinforcement Learning, Luo et al.
>
> [2.] Self-Improving Vision-Language-Action Models with Data Generation via Residual RL, Xiao et al.
>
> [3.] Planning for Proactive Assistance in Environments with Partial Observability, Kulkarni et al.

---

### Official Review · Reviewer_6drt · 2025-10-26

**Soundness:** 3
**Presentation:** 3
**Contribution:** 2
**Rating:** 2
**Confidence:** 5

**Summary:**

This paper investigates the problem of learning from human interventions in robotic control.

The authors observe that prior works assume humans intervene optimally and instantaneously based on per-step suboptimality, which is unrealistic.

Using data from human–robot interaction experiments, they find that real human interventions are suboptimal and non-Markovian, typically triggered by stagnation in task progress rather than instantaneous errors.

They propose

(1) a data-driven model of human intervention that predicts intervention timing based on progress stagnation over a short temporal horizon, and

(2) STEER a hybrid RL–IL algorithm that treats interventions as a signal to guide exploration rather than as expert demonstrations. STEER combines off-policy RL updates with a decaying-weight behavior cloning term on human corrections.

**Strengths:**

1. I like the empirical studies that showcase the characteristics of human intervention: human is not optimal, human is not deciding based on single step (non-Markovian). But I suspect if $1/k \log_{\gamma}\cfrac{V*(s_t)}{V^*(s_{t+k})}$ is a correct metric and if the result is valid as we don't know the expertise levles and the number of the human subjects involved in the experiment.

2. The paper proposes RL+IL method to learn from with human-in-the-loop data -- which is not novel.

**Weaknesses:**

1. **Critical experiment information is missing.**  See questions.
2. **Critical related works are missing**.
    * In [1], reward and human intervention are both used to learn from online human-in-the-loop data.
    * In [2], reward-free human-in-the-loop learning can achieve high learning efficiency and the paper finds that adding RL reward actually harms final performance.
    * In [3], BC loss is added to accelerate reward-free human-in-the-loop learning.
    * In [4], TD3 + BC is used as an effective offline RL method, which is highly similar to the Section 5
3. The contributions of this paper is questionable. The core algorithmic innovation is "BC regularized off-policy RL" which is already be implemented in reward-free human-in-the-loop learning[3] and offline RL[4]. And the empirical study on the characteristics of the human intervention is not convincing. Also, the connection between the empirical study in Sec 4 and the algorithm in Sec 5 is unclear.


[1] Efficient Learning of Safe Driving Policy via Human-AI Copilot Optimization

[2] Learning from Active Human Involvement through Proxy Value Propagation

[3] Data-Efficient Learning from Human Interventions for Mobile Robots

[4] A Minimalist Approach to Offline Reinforcement Learning

**Questions:**

### **Critical experiment information is missing.**

1. Is there a complete footage of the human subject experiments?
2. What's human subjects performance if doing full intervention?
3. What's their expertise in the task?
4. Is it possible that 3D mouse is not a good control device? Have you tried different control devices? Have you study the action distribution comparing "simulated human" and real human?
5. What if you have multiple human subjects with different expertise on the task? Does the analysis in Sec 4 still hold?
6. Did you get IRB?
7. How many repeat experiments are conducted with real human?
8. How to evaluate the trained policy? How many episodes are run? How diverse the training scenarios and the test scenarios are?



### **Section 4 The characteristics of human intervention**

I find this section interesting, but there are still many unresolved issues.

1. How to ensure that the optimal experts' $\pi^*$ and $V^*$ are optimal? Could you post the metrics for this expert policy in the experiment tables / figures?
2. Does Figure 3 plots all $Q^*(a_h)$ for all human actions during the training? Do you observe the Q values for human actions changes across the training? For example, when human becomes tired as the agent becomes performing, will the quality of human actions drop --- this explain why RL helps at the later stage of training?
3. Have you tried to use the action difference, or the probability of agent actions under the expert's action distribution, as the gated function to build the intervention predictor? How it performs?
4. **The Table 1 is concerning.** Flipping a coin will give you 0.5 recall on predicting human intervention. The precision is too low to 0.141, which means the gated function asks human intervention unnecessary. I don't think the intervention model is valid and thus all downstream experiments on the simulated human are not convincing.
5. **How to ensure that the expert policy satisfies the human intention?** For example, in the "Can" task, what if the human prefers to pick up the can higher in order to ensure safety but the expert RL policy is just optimal and use the shortest path? The single success rate and the sparse reward is not good enough to capture the human intentions --- this might explain why the Q* values of human doing bad but the proposed gating function is still so bad.


### **Experiment**

Experiments are very concerning.

1. According to Appendix B.2.5: "We use the same parameters across all experiments and report the best result on 3 seeds." **I don't think it's a correct practice to report best result over multiple repeated runs**.
2. According to Appendix B.2.1, "To accelerate convergence, we initialized training with 25 demonstration trajectories, fixed the initial positions of both the robot endeffector and goal objects across episodes, and restricted the action space to only end-effector position deltas and gripper commands."
    * What is the behavior cloning performance on these 25 demonstration? Given that there is no diversity in the environment, I suspect whether 25 episodes BC warmup can already achieve good performance.
    * If I understand correctly, **there is no difference between episodes in terms of initial position, objects, goals?** I believe this severely undermines the soundness of the experiment, as it becomes unclear whether the policy learns to react to observations or simply memorizes the data—especially given the BC baseline is not provided.
3. Important baselines are missing. This paper only compares HIL and RLIF, yet there are many important baselines should be included.
    * Behavior Cloning with human demonstration (with independently collected 15k data).
    * Behavior Cloning with expert RL policy (with 15k data, and 40k data, to have a fair comparison).
    * HG-DAgger
    * PVP
    * IWR


### The intuition

1. First, I can't see the connect between the discussion in Sec 4 (let's put aside that whether they are valid, even though I think the conclusion is reasonable (non-Markovion, suboptimal), but the results are not convincing) and the only technical innovation the BC+RL in Sec 5.
2. So STEER is basically HIL + BC loss.
3. The decay BC weights basically suggests we want human intervention to guide exploration and RL reward to guide later training. But I think the reverse idea is more practical in real-world setting. I think mastering basic skills via large scale pretraining and finetune the model to address compounding error and OOD actions in closed-loop rollout via human-in-the-loop is more interesting as it avoid human cognitive cost in providing basic-level demonstrations.

**Details Of Ethics Concerns:**

Important information on human subjects experiment, including how the human subjects are recruited and treated/compensated, how long the experiments last, how many repeated experiments are conducted with human subjects for each task, the expertise levels of human subjects, is not provided.

---

> ### Author Response · Authors · 2025-11-28
>
> We would like to thank the reviewer for their detailed comments and feedback about the paper. We include experiments and clarifications to the questions raised:
>
> - **Critical related works are missing.**
>
> We would like to thank the reviewer for bringing our attention to these additional works.  In Figure 10, we include PVP [2,3] as an additional baseline and show that as STEER outperforms the method when the interventions are sub-optimal (refer pt C of common response for details on additional baseline). While [4] introduces a TD3+BC loss to learn from offline demonstrations, the central contribution of our work is to analyse online human interventions and then expose the surprising effectiveness of a relatively simple way of learning from interventions. We have updated our manuscript with references to additional papers [1-4]. (L137/L151)
>
> - **Additional baselines:**
>
> Refer to pt B. of the common response.
>
> - **Real-Robot Validation**
> Refer to pt A. of the common response.
>
> ## Experimental Information
> - **Experimental videos**
>
> We include footage of real-robot experiments across the four baselines. Results in Section 7 and Table 2, show that even with a real robot task, STEER is effective at learning from human corrections.
>
> - **What's the human subjects' performance if doing a full intervention?**
>
> We observe in our experiments that humans achieve a success rate of ~80% over the course of training across different baselines, with all human participants consistently achieving success across multiple episodes.
>
> - **VR interventions**
>
> STEER makes no assumptions about intervention modality and works across diverse interfaces. In Figure 17, we compare STEER with a 3D spacemouse and with VR interventions on the Robomimic Lift task; in both cases, the policy converges efficiently to high success. Despite VR interventions being similarly noisy and sub-optimal, STEER remains robust, demonstrating the generalizability of our findings across modalities.
>
> - **Clarification on experiments with real humans**
>
> Our experiments for the intervention analysis in Section 4 have been collected across four participants. We provide an updated clarification on our experimental setup in Appendix B.2. For the real-human experiments in Section 6 to evaluate the intervention learning algorithms, we run experiments with five different human interveners and report the average evaluation curves across all of them. For the real-robot experiment in Section 7, we run it across one human, but are in the process of scaling it to additional experiments.
>
> - **IRB**
>
> Our experiments involved human participants interacting with a computer interface to intervene with the simulated robot, and these research procedures were approved by our university’s internal review board.
>
> - **Evaluation details**
>
> For our experiments across sim and real, evaluations are run across 20 episodes with randomized end-effector positions and randomized object positions (in real). We adopt the standard training approach in RL to train and evaluate under the same environment.
>
> ## Section 4: The characteristics of human intervention
>
> - **How to ensure that the optimal experts' $\pi^V^$ are optimal? ….**
>
> Thank you for pointing this out. We train a policy with RLPD [1] till convergence (~100k environment interactions) on the sparse task-reward to obtain $\pi^V^$. The policy achieves a 100% success rate, and we will include this detail in the paper.
>
> - **Q-value plots of human actions over training:**
>
> We observe that the human sub-optimality across training is unaffected by the time, as we observe a constant ratio between the optimal state-action value and the state-human intervention action value over time (see Figure 15). We hypothesize that the current algorithms and setup are designed to be short, where each human participant has a maximum interaction time of ~30 minutes, so this eliminates the human sub-optimality because of tiredness.
>
> - **Additional gating model based on action difference:**
>
>  In Figure 11, we also compare the performance of STEER across different intervention simulators, including a setting where the intervention gating model is based on the difference between the optimal and the agent action. We highlight that STEER makes minimal assumptions about human intervention style and optimality, and thus, efficiently learns from interventions as compared to other baselines.

---

> > ### Author Response · Authors · 2025-11-28
> >
> > - **Precision and Recall of the when-to-intervene model**
> >
> > Table 1 includes a random baseline, and our model achieves substantially higher precision and recall in predicting human interventions. Although a balanced dataset would allow a random coin flip to reach 0.5 precision, the real data is highly skewed towards not intervening; our method’s strong recall indicates fewer false negatives. Our objective is to propose a better model for benchmarking algorithms, so while not perfect, the proposed gating model provides a fairer evaluation of all learning methods. Our method achieves a much higher recall - indicating that the simulation model intervenes more than humans do (but not less), and is potentially an upper bound on human effort. We also show that our proposed gating model has a higher correlation with real world rankings of algorithms than other methods (see Figure 12).
> >
> > - **How to ensure that the expert policy satisfies the human intention?**
> >
> > In our experiments, we specify the task using the sparse reward, which doesn't capture any additional preferences the human interveners could have for the task, however, this could be extended with further reward engineering  or learning.
> >
> > ## Experimental setup
> >
> > - **According to Appendix B.2.5: "We use the same parameters across all experiments…**
> >
> > We apologize for the confusion; however, here we imply that we run a parameter sweep for all baselines and then report the mean result across 3 seeds for the best-performing hyperparameter for each baseline. We would update the manuscript to clarify this.
> >
> > - **According to Appendix B.2.1, "To accelerate convergence …., **
> >
> > In our simulated experiments, to accelerate convergence and reduce training time for multiple participants, the object’s initial position is fixed, but the end-effector’s initial position is randomized, requiring the policy to generalize across varying initial positions. In real experiments, we scale this to randomize the initial positions of the robot across training and evaluation as we conduct experiments with researchers, allowing longer experimental runs.
> >
> >
> > We hope that we have addressed all your concerns and would be happy to provide further clarifications and experimental results.

---

### Official Review · Reviewer_VURH · 2025-10-31

**Soundness:** 2
**Presentation:** 3
**Contribution:** 2
**Rating:** 2
**Confidence:** 4

**Summary:**

The paper does an analysis of human intervention in the setting of robot learning and proposes a method for learning from human interventions that takes into account what is the best way to use human intervention to learn a policy.

**Strengths:**

Analysis on human intervention is original:
The idea to analyze human interventions is novel. The paper collects real intervention signal and performs an analysis of when humans intervene. The analysis is informative and the proposed model takes into account this analysis for better performance.

The baselines are thorough:
The paper takes into account all of the relevant baselines.

**Weaknesses:**

Limited experimental evaluations:
The paper only evaluates on three simulated tasks from the same benchmark which is not enough. The baselines the method compares to all have real robot experiments to back up the methods, which in the setting of learning from human demonstrations is important. The sections of real world experiments is also misleading as it sounds like it is real robot experiments when it is only real human intervention.

Lack of details on human study:
The paper does not mention details of the human study, which is central to the analysis.

Generalization to different intervention styles:
The experiments is ran with two humans. Considering how different humans may have different intervention styles, more humans are needed to test the proposed model.

**Questions:**

What are the details of the human study? How many participants were there?

How would the method on a real robot, where the interventions might look a lot more different?

---

> ### Author Response · Authors · 2025-11-28
>
> We would like to thank the reviewer for taking the time to review our paper, highlighting our novel human analysis, and providing us with valuable feedback. We include additional experiments and comments in the updated manuscript and below to respond to their concerns.
>
> - **Real-Robot Validation**
>
> Refer to point A. of the common response.
>
> - **Human Study Details**
>
> In Appendix B.2, we include additional details about the human studies in our paper. We conducted experiments with four different participants for the experiments, across all algorithms and tasks. For the human analysis, we run the algorithm from HIL-SERL as our baseline to collect intervention data across four different users with different user styles and carefully understand when and how humans intervene. This further guided our improved benchmarking and STEER, which is more efficient compared to the baselines.
>
> - **Generalization to different intervention styles**
>
> We thank the reviewer for this comment. Following this, we increase the number of human interveners in Figure 6 to four participants, and include real robot experiments in Section 7, showing that the insights derived from the experiments on STEER hold across a larger pool of humans and on real-world systems, highlighting that STEER can learn efficiently from a wide range of intervention styles and strategies. In Appendix A.2, we include experiments with different gating mechanisms in simulation and show that variants of STEER are the best performing methods across all of them.

---

### Official Review · Reviewer_Yr8d · 2025-11-01

**Soundness:** 4
**Presentation:** 4
**Contribution:** 2
**Rating:** 6
**Confidence:** 4

**Summary:**

This paper presents a detailed analysis of how humans interact with robots in a human-in-the-loop policy learning setting. The authors study when and how humans intervene or take over while supervising an autonomous agent operating in an environment. Building on this analysis, they propose a new **gating mechanism** that predicts intervention timing based on the recent progress over a horizon. This mechanism is designed to simulate human intervention behavior more realistically. Evaluations using real human intervention data show that this gating model better captures human behavior than existing baselines.

Furthermore, motivated by the observation that human interventions can be suboptimal, which are evaluated via a critic from a trained RL policy, the authors propose a **learning-from-intervention** framework. This framework incorporates human interventions as a decaying auxiliary supervision loss during off-policy RL training, guiding exploration without assuming intervention optimality. The approach is evaluated against imitation learning (IL) and RL-based human-in-the-loop baselines.

**Strengths:**

- The paper is **well-written and easy to follow**, offering a detailed analysis of how and when people intervene in human-in-the-loop manipulation tasks.

- The proposed **gating mechanism** is intuitive and well-motivated, and its validation with real human data is convincing.

- The **learning-from-intervention framework** is compared against a range of baselines and evaluated across multiple dimensions such as human effort and final task success, showing careful experimental design.

**Weaknesses:**

- **Ablations on the RL vs. Supervision Trade-off**: It is unclear how much of the final performance gain comes from the RL component versus the intervention-based supervision. It would strengthen the work to include ablations such as: (i) training only with sparse rewards, and (ii) using only the corrected segments for behavioral cloning without decay (still distinct from HG-DAgger since no offline dataset is used).

- **Intervention Modality**: In this work, all interventions are provided via a space mouse, which raises questions about the role of interface modality. The interventions appear to be non-Markovian (e.g., humans waiting for clear mistakes due to the cost of correction). It is unclear whether the proposed method would be as effective on intervention learning with kinesthetic teaching or VR teleoperation.

- **Real-World Validation**: While the proposed framework is tested with real human interventions in simulation, experiments on a physical robot would make the contribution significantly stronger, especially given the practical nature of human-in-the-loop policy learning.

**Questions:**

- **Effectiveness with Other Intervention Types**: Would similar intervention patterns emerge with more intuitive interfaces like kinesthetic teaching or VR teleoperation? Some discussion on this would be valuable.

- **Baselines for Reactive Interventions**: The paper’s analysis suggests that human interventions are largely reactive (i.e., following temporary policy failures). However, the comparisons against supervised learning methods mainly focus on HG-DAgger. Other methods, such as Sirius (Liu et al. 2023) [1], also model reactive supervision and would serve as more appropriate baselines. Why are only RL-based methods evaluated with real human interventions in the performance plots?

Overall, I found this to be a well-motivated paper that tackles an important aspect of human-in-the-loop learning. I appreciate the authors’ clear presentation and the effort put into both analysis and experimentation. I would be happy to hear the authors’ responses and clarifications to the questions and points raised above, as they will help me better understand the scope and impact of this work.

[1] Liu, Huihan, et al. ‘Robot Learning on the Job: Human-in-the-Loop Autonomy and Learning During Deployment’. Robotics: Science and Systems (RSS), 2023.

---

> ### Author Response · Authors · 2025-11-28
>
> We thank the reviewer for their helpful feedback and review of our paper, emphasizing our experimental setup and analysis. We respond to the above concerns below :
>
> - **Real-Robot Validation**
>
> Refer to point A. of the common response.
>
> - **Ablations on the RL vs. Supervision Trade-off**
>
> In Figure 18, we show HG-Dagger with real human participants without any offline pretraining. STEER still outperforms this baseline by using interventions for fast supervised updates and off-policy RL, and HG-Dagger without demos is lower in performance to the demo-initialized baseline, underscoring the value of both offline demos and online interventions.
>
> - **VR interventions**
>
> STEER makes no assumptions about intervention modality and works across diverse interfaces. In Figure 17, we compare STEER with a 3D spacemouse and with VR interventions on the Robomimic Lift task; in both cases, the policy converges efficiently to high success. Despite VR interventions being similarly noisy and sub-optimal, STEER remains robust, demonstrating the generalizability of our findings across modalities.
>
> - **Baselines for Reactive Interventions:**
>
> Refer pt 3. of the common response.
>
> We hope that the above experiments and comments answer the reviewers' concerns. We would be happy to provide additional results and clarifications to further strengthen our contribution.

---

### Author Response · Authors · 2025-11-28
**Common Response**

We would like to thank all the reviewers for their feedback and comments. We summarise the updates to our experiments and manuscript below:

- **Real Robot Experiment**

In Section 7, we include experiments with a Franka robot for a pen-in-bowl task. We observe that STEER converges to ~50% more success with 2x fewer interventions given the same environment interaction budget. Notably, it achieves a higher performance with 2x less environment interactions. We ablate the decay component in STEER, and show that accounting for the policy-human divergence (Figure 4), enables STEER to converge to ~70% success rate in 5k environment interactions . Thus, our findings, which involve a real robot and human intervention, align with our observations across multiple simulated experiments involving both real and simulated humans. We include the complete footage of real robot experiments in the supplementary material (Additional details in Section 7 of the updated paper).

- **Connection between the human analysis and STEER**

Overall, we present two contributions in the paper: 1. Build a better proxy for simulating human interventions (from real human intervention data), thus enabling more reliable benchmarking for developing algorithms 2. And then building on insights, propose a more efficient and robust algorithm (STEER) to learn from human corrections, which we validate across the simulated benchmark, and real human participants intervening on a simulated and real robot.

The analysis showing the non-Markovian and sub-optimal nature of human interventions primarily motivates a better gating model for more reliable benchmarking of all human intervention learning algorithms. We show in Section 4 that our proposed model achieves higher precision and recall when modelling real human intervention data. We observe that the decision of “when” to intervene is non-Markovian, i.e., dependent on the past horizon of observations.  Additionally, in Figure 3, we show that humans and the robot policy diverge over the course of training, which motivates the decay parameter in STEER that reduces the supervision to stale/old human corrections. This is essential for efficiently learning from human interventions as observed in Table 2 and Figure 10. Thus, the human analysis independently serves to guide us in building better algorithms and a benchmarking platform for learning from interventions.

- **Additional baselines**

In Appendix A.1, based on the feedback, we include several additional baselines 1. PVP [1] - a method that augments TD-loss with additional supervision via the interventions, 2. SIRIUS and IWR [2-3] - behavior cloning-based baselines with additional weighting mechanisms for the intervention and non-intervention transitions. In Figure 10, we observe that STEER is the best-performing policy across all settings. A major benefit of STEER over other baselines is that it makes minimal assumptions about the quality and pattern of human interventions and, as a result, can leverage a wide range of interventions with different levels of optimality. In our simulated intervention and real human intervention experiments, STEER is robust to the suboptimal corrections and converges to a higher success rate much more sample efficiently (both environment steps and human intervention effort) than the baselines.

- **Simulation experiments across multiple intervention models**

In Appendix A.2, we run all the baseline learning from intervention algorithms with all the gating models in Section 4, and an additional gating model that takes into account the action difference between the expert and the agent, based on reviewer feedback. To quantify how well each simulation signal predicts real-robot performance, we compute the Spearman rank correlation between the real-robot ranking and the ranking induced by each intervention model (see Table 3). In Figure 12, the stagnation-based gating model achieves the highest correlation with the real-robot ordering, indicating that it is the most reliable proxy for real-world performance.


- **Intervention learning with perfect experts**

In our simulated experiments, human sub-optimality is mirrored via sub-optimal RL checkpoints. In Figure 13, we compare the performance of all methods with perfect experts (RL-trained experts that are easy to imitate) and observe that most baselines achieve significant improvements in performance with close to optimal interventions, further emphasizing that STEER does not need the interventions to be optimal and outperforms the baselines under practical, noisy conditions.

---

> ### Author Response · Authors · 2025-11-28
>
> - **Scaling human interveners in sim experiments**
>
> Based on feedback from the reviewers, we scale the number of human participants in our real intervention experiments to four to test the generalisation of our insights across multiple participants. In the updated Figures 5 and 6, we observe that STEER consistently has higher performance and requires lower human effort as compared to the other intervention learning baselines.
>
> [1] Learning from Active Human Involvement through Proxy Value Propagation, Peng et al.
>
> [2] Robot Learning on the Job: Human-in-the-Loop Autonomy and Learning During Deployment, Liu et al.
>
> [3] Human-in-the-Loop Imitation Learning using Remote Teleoperation, Mandlekar et al.

---

### Note · Authors · 2026-01-24

I have read and agree with the venue's withdrawal policy on behalf of myself and my co-authors.